# SWE-Mirror: Scaling Issue Resolving Datasets by Mirroring Issues Across Repositories

## Abstract

Creating large-scale verifiable training datasets for issue-resolving tasks is a critical yet notoriously difficult challenge. Existing methods on automating the Gym environment setup process for real-world issues suffer from low success rates and high overhead. Meanwhile, synthesizing new tasks within existing Gym environments leaves the vast pool of real-world issue-resolving history untapped. To maximize the utilization of existing Gym environments and also the rich data of issue-resolving history on GitHub, we introduce SWE-Mirror, a pipeline that distills a real-world issue's semantic essence, mirrors it into another repository with a configured Gym environment, and re-animates it as a verifiable issue-resolving task. SWE-Mirror reuses existing Gym environments along with the vast pool of issue-resolving history hosted on GitHub to construct a large-scale dataset of *mirrored* authentic and verifiable tasks. Applying SWE-Mirror to 40 repositories across 4 languages, we have curated a dataset with 60,671 issue-resolving tasks and demonstrated the value of our dataset by training and evaluating coding agents at various scale. Post-training experiments show that models trained with the dataset exhibit improvements in issue-resolving capabilities. Among Qwen2.5-Coder-Instruct based models, we established a new state-of-the-art (SOTA) by extending the dataset size to over 12,000 high-quality trajectories on the OpenHands agent framework, which increases the resolve rate on SWE-Bench-Verified by **+21.8%** for the 7B model and **+46.0%** for the 32B model and validates the effectiveness of our approach.

## 1 Introduction

Large Language Models (LLMs) have demonstrated remarkable capabilities in various code generation tasks (Chen et al., 2021; Austin et al., 2021; Liu et al., 2023; 2024; Jain et al., 2024; Li et al., 2022; Luo et al., 2025; Guo et al., 2024; Wan et al., 2025), fundamentally reshaping the landscape of software development. As the research community broadens its focus to more complex and real-world challenges (Zhang et al., 2024a;b; Jiang et al., 2025), resolving real-world issues has emerged as a critical frontier (Jimenez et al., 2024; OpenAI, 2024; Zan et al., 2025; Wei et al., 2025). A verifiable issue-resolving task instance, exemplified by benchmarks like SWE-Bench (Jimenez et al., 2024; OpenAI, 2024; Yang et al., 2025b), consists of two primary components:

- **Task Context**: This includes the issue with related pull-request(*i.e.*, PR) and the corresponding repository snapshot(*i.e.*, CodeBase). Normally we can get a problem statement detailing a specific issue (*e.g.*, a bug report or feature request) as the task description, and reference patches for validation and ground-truth.

- **Gym**: This is an executable environment equipped with validation harness, including test commands and log parsers to verify proposed solutions and provide reward for training.

A severe imbalance exists (Pan et al., 2025; Badertdinov et al., 2025) in the effort required to acquire these two components. While Task Contexts can be gathered from platforms like GitHub with relative ease, engineering a functional Gym is a significant bottleneck, demanding meticulous and often unscalable manual effort (Jimenez et al., 2024; Zan et al., 2025; Pan et al., 2025). This difficulty arises because a universal, one-fits-all Gym is infeasible in the diverse software ecosystem. Each repository—and often, each specific version—requires a unique configuration of dependencies, build processes, and testing frameworks. Consequently, the immense effort invested in creating a

single Gym typically supports only one specific task or, at best, a small cluster of closely related ones. This reality forges a rigid **one-to-one dependency** between Task Context and Gym, posing a fundamental barrier to scaling up the issue-resolving datasets.

Faced with this scaling challenge, the research community has pursued two orthogonal approaches to scaling the issue-resolving dataset for training: ❶ **Scaling tasks via synthesizing problems.** This approach maximizes the utility of Gyms by synthesizing new tasks that are compatible with them. Works like SWE-smith (Yang et al., 2025b), R2E-Gym (Jain et al., 2025) and SWE-Synth (Pham et al., 2025) programmatically mutate or rewrite repositories' components to inject bugs and generate a large volume of artificial tasks. ❷ **Scaling tasks via setting up Gyms.** This orthogonal approach confronts the Gym creation bottleneck directly by attempting to automate the setup process (Badertdinov et al., 2025).

While both approaches offer paths to scale, they present a difficult trade-off. The synthesis approach achieves scale but generates problems that are artificially created, failing to leverage the vast and history of authentic software evolution found on platforms like GitHub—the very source of problems this research field aims to solve. Conversely, the Gym automation approach engages with this real-world data but faces significant engineering hurdles. The success rate of automatically configuration remains low, and incurs staggering storage costs. With each Gym environment consuming approximately 1GB, scaling to 100,000 instances would demand a 100 Terabytes of storage.

This presents the community with an untenable choice: pursue scalability with tasks disconnected from rich source of real-world software evolution, or engage with authentic data at a prohibitive engineering and storage cost. This dilemma leads to a research question:

> *How can we leverage the vast and ever-growing history of software evolution on GitHub using only a small, manageable set of reusable Gyms?*

To answer this question, we must break the **one-to-one dependency** between the Task Context and the Gym. Our approach involves hosting an issue-resolving task from one repository within a pre-existing Gym configured for another. We draw inspiration from research on **issue mirroring** (Guan et al., 2025), which observes that programs with analogous functionalities often share analogous bugs and features. While prior work has leveraged this insight to *find* bugs across similar frameworks (*e.g.*, PyTorch[1] and TensorFlow[2]), we propose to significantly extend this idea to programmatically *mirror* them—re-instantiating a PR from a source project into a target project to create a new task. Observations supporting the feasibility can be summarized as follows:

1. *Shared Analogous Components:* Similar projects often share analogous components rooted in common architectural patterns, dependencies and APIs and may suffer similar issues.
2. *Portable Problem Logic:* Software issues often encapsulates core logical problem that can be abstracted from its original context and can re-instantiated within a similar project.
3. *Transferable Validation:* Issues from a repository is typically accompanied by a validation mechanism (*e.g.*, a test case that fails before the fix and passes after). which can be adapted and transferred to the target repository to verify the successful replication of the issue.

To this end, we introduce SWE-MIRROR, a pipeline that systematically mirrors real-world PRs and issues from a source repository in the wild into a functionally similar target repository which has a configured Gym. By breaking the **one-to-one dependency** between Task Context and Gym, SWE-MIRROR dramatically multiplies the available tasks of any single Gym and unlocking a vast pool of authentic issue-resolving histories. The main contributions of this paper are summarized as follows:

❶ **Technique:** We propose SWE-MIRROR, a novel paradigm and methodology for scaling issue-resolving datasets by mirroring real-world issues across repository.

❷ **Large-Scale Dataset:** We release SWE-MIRROR-60K, a large-scale dataset containing over 60,000 verifiable tasks. These tasks are composed of authentic issues mirrored into a small set of robust Gyms. A comparison with other datasets is shown in Table 1.

❸ **Empirical Validation and Methodology:** We conduct extensive experiments exploring various agentic posttraining methods on SWE-MIRROR-60K. Our results not only demonstrate

---

[1]https://pytorch.org

[2]https://www.tensorflow.org

that finetuned models achieve significant performance gains on SWE-bench Verified (OpenAI, 2024) and Multi-SWE-bench-Flash (Zan et al., 2025), but also provide insights into training strategies for this domain. We also provide strong empirical evidence for scaling law (Kaplan et al., 2020) of dataset size in software engineering tasks.

Table 1: Comparison of SWE-MIRROR with other issue-resolving datasets. The symbols indicate whether a dataset possesses the feature (✓), lacks it (✗), or possesses it partially (✓).

| Dataset | #Tasks | # Repos | Hidden Tests? | Verifiable? |
|---|---|---|---|---|
| SWE-rebench (Badertdinov et al., 2025) | 20k | 2k | ✓ | ✓ |
| SWE-Gym (Pan et al., 2025) | 2.4k | 11 | ✓ | ✓ |
| SWE-Fixer (Xie et al., 2025) | 110k | 856 | ✓ | ✗ |
| SWE-Smith (Yang et al., 2025b) | 50k | 128 | ✗ | ✓ |
| **SWE-MIRROR-60K (Ours)** | **60k** | 40 | ✓ | ✓ |

## 2 METHODOLOGY

As illustrated in Figure 1, this process is structured as a three-phase pipeline: *(1) Task Collection*, where we collect high-quality and mirror-able real-world issues from GitHub; *(2) Task Mirroring*, where we mirror these issues into target codebases; and *(3) Task Verification*, which validates the integrity of the mirrored task instances. Worth-noting, SWE-MIRROR is an orthogonal method on scaling dataset to prior efforts working on setting up Gyms for SWE instances. Due to the limit of resources and time, we select Gyms for newest issue from SWE-Gym (Pan et al., 2025), SWE-rebench (Badertdinov et al., 2025) and Multi-SWE-RL (Zan et al., 2025), and set the time limit of running the *whole* test suites to 5 minutes and the memory limit to 1GB. In addition, we also perform basic functional check of each Gym via running all test suites and check the output manually.

### 2.1 PHASE 1: TASK COLLECTION

The objective of this initial phase is to source a pool of potentially mirror-able issues for each target CodeBase with existing Gym. Given the vast volume of issues on GitHub, we employ a two-stage search strategy to narrow the candidate pool to a manageable scope. For a given CodeBase, we first leverage QWEN3-32B (Yang et al., 2025a) to analyze its README file and generate five descriptive keywords. Using the GitHub REST API[3], we then search for repositories using these keywords as query, retrieving the top 20 repositories ranked by stars and issue counts. Subsequently, we collect all pull-requests and linked issues from these candidate repositories and apply a filtering process, using a combination of hand-crafted rules and LM-based heuristic to identify *high-quality* and *mirror-able* issues. We expand the rules and LM-based heuristic in Section A.1.

### 2.2 PHASE 2: TASK MIRRORING

The objective of this phase is to mirror the candidate issues into their designated target Gyms. The process begins by employing GPT-4O-2024-0513 (OpenAI, 2024) to distill the related functionality, core logic, current and expected behavior and observable symptoms of a source issue into a concise *abstract description* which serves as a primary input for our three-step mirroring workflow with GPT-4.1 (OpenAI, 2025) as the backbone LM:

- **Mirroring Validation**: The primary goal of this initial step is to establish a concrete, executable contract that formally defines what constitutes a correct resolution of the issue-resolving task. An agent referred as **Test Agent**, prompted with the *abstract description*, is responsible for generate a new test case within the target Gym's existing test suite. Those tests are designed to *pass* under the current codebase state, but will *fail* once the next step introduced the issue successfully. The output of this step is the *test.patch*. This patch serves a dual purpose: it acts as a precise guide for the next step and, ultimately, as the hidden tests for evaluating the correctness of submissions from coding agents.

---

[3]https://docs.github.com/en/rest

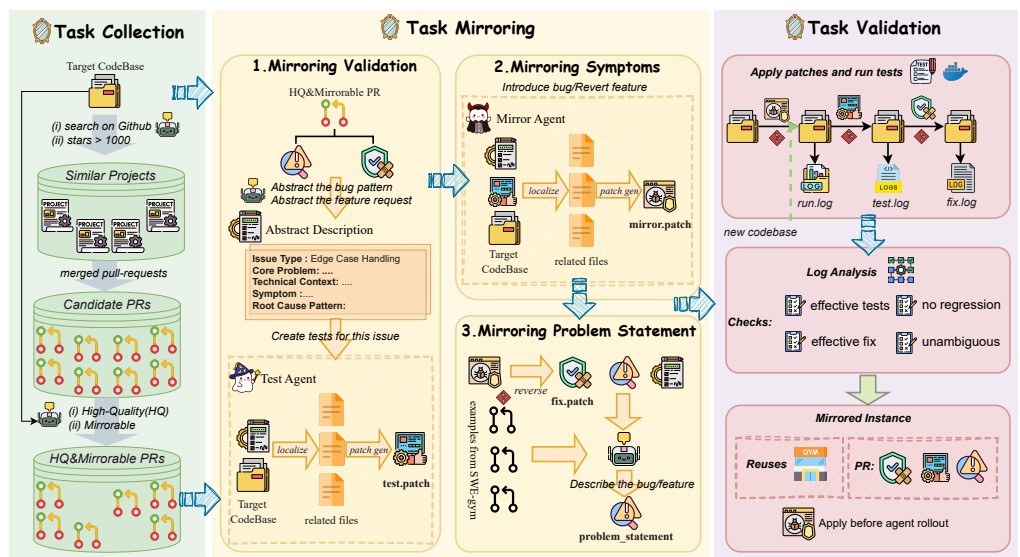

Figure 1: Overview of SWE-MIRROR pipeline.

- **Mirroring Symptom**: With the validation tests established by the test.patch, this second step aims to *re-animate* the issue within the target repository's code. A different agent-**Mirror Agent** takes the *abstract description* for semantic context and the file paths and function names from the *test.patch* as a strong structural prior. Its objective is to surgically modify the application's source code to specifically cause the new test case to fail. The resulting modification, packaged as the *mirror.patch* after removing comments, becomes the starting point of the task. We also programmatically create its inverse, the *fix.patch*, which serves as a reference solution.
- **Mirroring Problem Statement**: This final step is responsible for synthesizing a natural-language problem description that will be presented to the coding agents. The goal is to create a description that is not only accurate to the mirrored bug but also feels native. To achieve this, the LM is prompted with a rich set of context following the quality criteria described in SWE-bench-Verified (OpenAI, 2024), including: (1) the original GitHub issue description for semantic context; (2) the generated *test.patch* and the *fix.patch* to ground the description in the specific files and functions of the target codebase; and (3) few-shot examples of other issues from the SWE-Gym to ensure stylistic consistency. The resulting *problem_statement* synthesizes these inputs into a self-contained description.

The successful execution of this workflow yields a final mirrored task. Each task instance is a self-contained data structure containing the following fields:

- *mirror.patch*: A patch that introduces a bug or reverts a feature in the codebase. Applying this patch creates the starting point of the issue-resolving task.
- *test.patch*: A patch used to test the correctness of a submission, in line with benchmarks like (Multi-)SWE-Bench (Jimenez et al., 2024; Zan et al., 2025; OpenAI, 2024). This should not be revealed to the coding agent system.
- *fix.patch*: Reference solution for the task, created by reversing the *mirror.patch*.
- *problem_statement*: Task description presented to coding agents in natural language.

Detailed workflow design, prompts used in this phase are demonstrated in Section A.2.

## 2.3 PHASE 3: TASK VERIFICATION

In this Phase, we first perform a sanity check to ensure all patches can be applied without error. Concretely, the *mirror.patch* can be applied to the *base_commit* of the original code base, the *test.patch* and *fix.patch* should be appliable after the application of *mirror.patch*. Then we conduct an execution-based validation, executing the full test suite under three states.

1. *Run.log*: Run all tests after apply *mirror.patch*.
2. *Test.log*: Run all tests after apply *mirror.patch* and *test.patch*.
3. *Fix.log*: Run all tests after apply all three patches.

Following Multi-SWE-bench (Zan et al., 2025), we analyze the test status transitions across these logs and apply strict filtering rules to accept only unambiguously correct mirrored tasks:

1. **Effective Tests:** the application of *test.patch* should introducing new tests without affecting existing tests. Comparing test status in *Run.log* and *Test.log*. Only PASSED→PASSED, FAILED→FAILED SKIPED→SKIPED, and NONE→FAILED are permitted.

2. **Effective Fix:** The *fix.patch* must fixes somethings. So comparing status in three logs, least one test with ANY→FAILED→PASSED transition is required.

3. **No Regressions:** No test may exhibit a transition that indicates the fix introduced a new bug, so transitions in PASSED→PASSED→FAILED and SKIPPED→SKIPPED→FAILED are not allowed.

4. **No Flaky Tests:** Instances with flaky tests are discarded, detected with multi-runs.

Only instances that pass this rigorous validation are included in our final dataset. We perform detailed framework analysis in Section A.3 which studies ❶ How effective is our LM-based pre- filter? ❷ What is the end-to-end mirror success rate for promising candidates? and ❸ Are the final mirrored tasks semantically consistent with the original issues and seems realistic?

## 2.4 DATASET STATICS AND FEATURES

We apply SWE-MIRROR on 40 repositories across 4 language. Since we enable sampling in Section 2.2, we can sometimes get more than one mirroring results, we perform deduplication to ensure that every instance have different F2P tests and each *fix.patch* modifies different content of the code base. The final dataset comprises 60,671 validated tasks. Table 2 presents a detailed statistical overview of the SWE-MIRROR-60K.

| | **Repos** | **Instances** | **Fix patches** | | **Unit tests** | |
|---|---|---|---|---|---|---|
| Language | #Num | #Num | #Hunks | #Lines | #P2P | #F2P |
| Python | 31 | 46,820 | 3.0 | 38.5 | 1,025.8 | 31.2 |
| Rust | 6 | 7,183 | 2.4 | 36.8 | 627.3 | 80.2 |
| Go | 2 | 4,056 | 3.3 | 42.5 | 107.1 | 7.5 |
| JavaScript | 1 | 2,612 | 2.7 | 36.2 | 216.0 | 33.8 |

Table 2: Dataset stastics of SWE-MIRROR-60K

## 3 EXPERIMENTS

In this section, we present a comprehensive empirical evaluation of our approach. We first detail the experimental setup, including our agent framework, data collection process, and post-training methodology. We then present the main results on two challenging benchmarks, demonstrating that our datasets boost the performance of base models. Finally, we conduct in-depth ablation studies to analyze the impact of data scale, training strategies, and the generalization of multi-lingual training.

## 3.1 EXPERIMENTAL SETUP

**Agent Scaffolding.** We selected OpenHands (Wang et al., 2025), an open-source, event-driven platform, as the agent framework for all experiments. OpenHands enables LLM agents to iteratively edit files, execute shell commands, and browse the web within sandboxed containers. This framework is known for establishing strong and reproducible baselines on benchmarks like SWE-Bench. For our experiments, we equipped the agent with 3 tools: *str-replace-editor* for file editing and reading, *execute-bash* for command execution and *finish* to stop and submission. We use MOpenHands[4] for languages other than Python as the agent scaffold.

---

[4] https://github.com/multi-SWE-Bench/MopenHands

**Agent Trajectory Collection.** To generate training data, we employed high-performing expert LLMs (*Claude-3.7-Sonnet* and *Claude-4-Sonnet*) to produce agent trajectories on a 15k subset of our SWE-MIRROR-60K dataset. For each task, we executed 3 trials with a temperature of 1.0 and a maximum of 100 rounds. A trajectory was considered successful only if ❶ it ends with a *finish* action and ❷ the set of tests passed after applying the submitted patch are a superset of the tests fixed by the ground-truth patch. This rigorous process filtered out 6,431 successful and high quality trajectories. We combined these with 6,025 trajectories from prior experiments on SWE-rebench (Badertdinov et al., 2025), creating a final post-training dataset of 12,456 trajectories.

**Agentic Post-training.** We use QWEN2.5-CODER-INSTRUCT-7B (QWEN ET AL., 2025) and 32B models as our base, resulting in our final models, SWE-MIRROR-LM-7B and SWE-MIRROR-LM-32B. The models were trained for maximum 3 epochs. We utilized AdamW (Loshchilov & Hutter, 2019) optimizer with weight decay of 0.01 and cosine learning rate schedule with warmup ratio of 0.1, peaking at learning rate of 5e-5. Specifically, our loss masking technique ensures that the loss is computed only for valid assistant turns that result in well-formed actions, a strategy we analyze in detail in Section 3.3. For experiments involving trajectories less than 4k, we set maximum learning rate as 1e-4 and trained 5 epochs using trajectories only from SWE-MIRROR-60K.

**Evaluation Benchmarks and Metrics** We evaluate our models on two primary benchmarks. The first, SWE-Bench-Verified (Jimenez et al., 2024; OpenAI, 2024), is a high-quality, human-curated set of 500 real-world software engineering issues in Python. The second, Multi-SWE-Bench-Flash (Zan et al., 2025), is a benchmark of 300 tasks designed for rapid evaluation of multi-lingual generalization capabilities. Performance is measured by the *Resolved Rate (%)*, which is the percentage of tasks solved successfully. Key hyperparameters were set as follows: the inference temperature was fixed at 0 for all experiments. The models were trained using a context length of 32,768. For evaluation our model in Table 3, we extended the context length to 131,072 with yarn and allowed for a maximum of 100 interaction rounds. For the ablation studies, we used a context length of 32,768 and a maximum of 100 rounds, but keep the model's only the last 5 observations' content from environment in the context. All evaluation scores are reported as pass1, with no test-time scaling.

## 3.2 EXPERIMENT RESULTS

Our main experimental results presented in Table 3 demonstrate the effectiveness of our approach. On the challenging SWE-Bench-Verified benchmark, our SWE-MIRROR-LM-32B achieves a resolve rate of **52.2%**, matching the performance of much larger models like DEEPSEEK-R1 and GPT-4.1 under the same agent framework. Furthermore, on Multi-SWE-Bench-Flash our SWE-MIRROR-LM-32B achieves score of **21.33%**, outperforming both DEEPSEEK-R1 and GPT-4.1. These results validate that training on a large-scale dataset of mirrored, real-world issues significantly enhances an model's abilities on agentic coding tasks.

## 3.3 ABLATION STUDIES

To dissect the key components contributing to our model's performance, we conduct a series of ablation studies designed to answer four fundamental questions. ❶ What are the effects of data scale and the training strategy used to handle errors within demonstration trajectories? ❷ How to better utilize the trajectories from expert model? ❸ does training enable the model to generalize across programming languages? ❹ Is the quality of tasks in SWE-MIRROR-60K comparable to real-world tasks? These experiments validate our core design choices regarding the dataset and training methodology and offer valuable insights for future work in agentic post-training for coding.

### 3.3.1 IMPACT OF DATA SCALE AND TRAINING STRATEGY

A fundamental challenge in training agents from demonstrations is how to handle intermediate error steps within otherwise successful trajectories. Expert-generated trajectories are not always monotonic paths to success; they often contain erroneous actions (*e.g.*, invalid function calls, incorrect arguments) that the expert subsequently self-corrects. Our guiding hypothesis is that training should

---

[5]https://github.com/multi-swe-bench/MopenHands

Table 3: Performance on SWE-Bench-Verified(SWE-V) and Multi-SWE-Bench-Flash (MSWE-Flash). The primary metric is *Resolved Rate (%)*. For Multi-SWE-Bench-Flash evaluation, we use MOpenHands[5], the multi language version of OpenHands.

| Model | Scaffold | SWE-V | MSWE-Flash |
|---|---|---|---|
| *Proprietary Models* | | | |
| GPT-4.1-0414 (OpenAI, 2025) | OpenHands | 57.6 | 14.33 |
| Claude-4-Sonnet (Anthropic, 2025) | SWE-Agent | 66.6 | – |
| | OpenHands | 70.4 | 25.00 |
| *Open-Source Models* | | | |
| Qwen2.5-Coder-Instruct-7B (Yang et al., 2025a) | OpenHands | 1.0 | 0.33 |
| SWE-agent-LM-7B (Yang et al., 2025b) | SWE-Agent | 15.2 | – |
| Qwen2.5-Coder-Instruct-32B (Yang et al., 2025a) | OpenHands | 6.2 | 0.67 |
| SWE-gym-32B (Pan et al., 2025) | OpenHands | 20.6 | – |
| SWE-agent-LM-32B (Yang et al., 2025b) | SWE-Agent | 40.2 | – |
| DeepSWE-32B-Preview (AI, 2025) | OpenHands | 42.2 | – |
| Skywork-SWE-32B (Zeng et al., 2025) | OpenHands | 47.9 | – |
| SWE-fixer-72B (Xie et al., 2025) | SWE-Fixer | 32.8 | – |
| Lingma-SWE-GPT-72B (Ma et al., 2024) | SWE-Syninfer | 32.8 | – |
| DeepSeek-R1-0528 (DeepSeek-AI et al., 2025) | OpenHands | 45.6 | 15.33 |
| Qwen3-Coder (Yang et al., 2025a) | OpenHands | 69.6 | 27.00 |
| *Ours* | | | |
| **SWE-Mirror-LM-7B** | OpenHands | **22.8** | **6.33** |
| **SWE-Mirror-LM-32B** | OpenHands | **52.2** | **21.33** |

focus gradient updates on generating valid, productive actions rather than replicating an expert's mistakes. This approach should not only prevent the model from learning to make errors but also improve its ability to recover from them.

To systematically answer this question, we designed and compared three strategies, each embodying a different hypothesis about the role of errors in learning:

- **Response Only:** This standard approach fine-tunes the model on all expert responses, including those that lead to errors. It risks teaching the model to replicate the expert's mistakes.

- **Error Pruning:** This strategy posits that error steps are detrimental and removes any error turn. While this avoids reinforcing mistakes, it comes at the high cost of discarding the context of how an agent recovers from an error, thereby losing learning opportunity for self-correction.

- **Error Masking:** This strategy, which embodies our central hypothesis, preserves the full trajectory context but surgically masks the loss on erroneous agent responses. This allows the model to learn from the context of a mistake without learning to make the mistake. By applying all gradient updates to valid actions, this method provides a rich learning signal for both action generation and error recovery.

Figure 2 plots the resolve rate on SWE-Bench-Verified as a function of the number of trajectories from SWE-MIRROR-60K. The results validate the quality of dataset and reveal two observations:

**Observation ❶: Model performance scales strongly with the amount of training trajectories.** For both model sizes and across all strategies, performance consistently improves as the number of training data increases (Kaplan et al., 2020). The 32B model trained with our *Error Masking* strategy improves its resolve rate from a baseline of 6.2% to 35.6% when trained on 4096 trajectories. This demonstrates a direct and powerful correlation between data volume and issue-resolving capability.

**Observation ❷: *Error Masking* consistently outperforms other training methods.** The performance gap between *Error Masking* and the other methods widens as the dataset grows, suggesting that the benefits of its richer learning signal compound with more data. By observing the entire sequence, the model learns how to recover from error states—a crucial skill that is lost when imperfect data is pruned. This makes *Error Masking* a more data-efficient and effective approach.

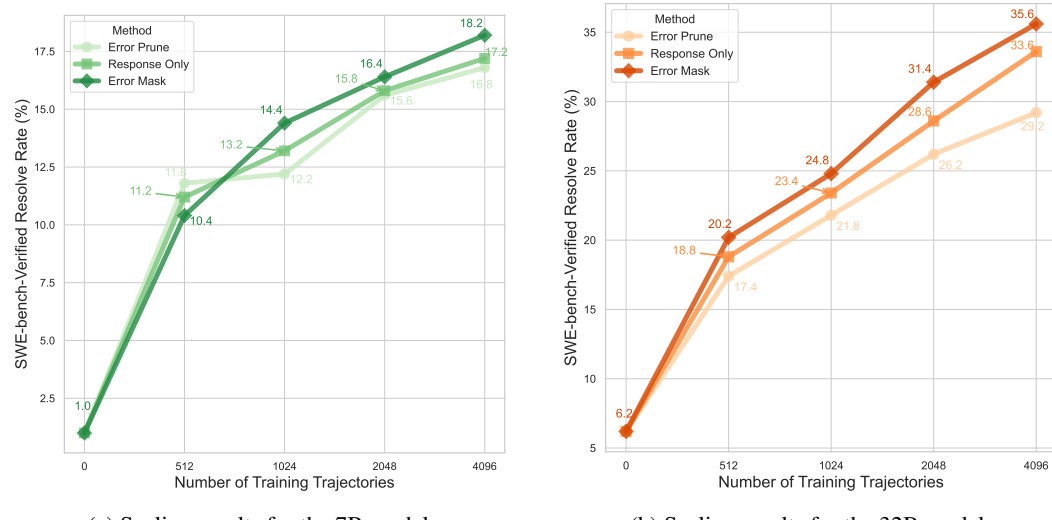

(a) Scaling results for the 7B model.  (b) Scaling results for the 32B model.

Figure 2: Performance on SWE-Bench-Verified as a function of training data scale for our three different training strategies. The *Error Masking* approach consistently outperforms the other methods.

### 3.3.2 CROSS-LINGUAL GENERALIZATION

To quantify the benefit of our multi-lingual dataset, we evaluated whether non-Python data could improve performance on the Python-only SWE-Bench Verified benchmark. We trained the 7B model on several monolingual subsets of our data (512 trajectories each) using the *Error Masking* recipe.

**Observation ❸: The model demonstrates strong cross-lingual generalization from non-Python data to Python tasks..** The results as presented in Table 4 shows that the 7B model trained exclusively on non-Python data still achieves a notable resolve rate on Python tasks. This provides strong evidence of **cross-lingual generalization**, wherein the model learns abstract problem-solving patterns and code semantics that transfer across languages. Notably, the model trained on Rust data yielded the most significant performance gain, which we attribute to the language's complexity and rich type system fostering more robust reasoning capabilities.

### 3.4 SYNTHETIC V.S. REAL ISSUE-REOLVING TASKS

To better investigate the quality of the task instances synthesized by SWE-MIRROR, we compare models trained on our synthetic data against those trained on real-world data. We trained the 7B model on 512 trajectories from SWE-REBENCH (real-world tasks) and 512 Python trajectories from our synthetic dataset, SWE-MIRROR-60K.

**Observation ❹: Synthetic data quality is competitive with real data**. The results presented in Table 4 show that the model trained on our synthetically generated Python data (SWE-MIRROR) achieves highly competitive performance. Specifically, the model trained on synthetic data reaches a resolve rate of 10.8%, which is remarkably close to the 11.4% achieved by the model trained on real-world trajectories from SWE-REBENCH. This demonstrates that our synthetic data generation process produces training signals of a quality and effectiveness comparable to those derived from real-world issue-resolving tasks, validating it as a scalable method for creating training data.

## 4 RELATED WORK

**Coding Agents.** Recent advancements in Software Engineering have spurred the development of agents capable of resolving real-world issues in repositories. These agents are evaluated on benchmarks like SWE-bench (Jimenez et al., 2024) and Multi-SWE-bench (Zan et al., 2025). A significant body of work focuses on agent design. For instance, OpenHands (Wang et al., 2025) introduces an event-driven platform that empowers LLM agents to iteratively edit files and execute commands. SWE-Agent (Yang et al., 2024) introduces Agent-Computer Interface (ACI) to provide LLM agents

| Training Language | Resolve Rate (%) | Improvement (%) |
|---|---|---|
| Base model | 1.0 | – |
| + Go | 10.2 | ↑ 9.2 |
| + Rust | 11.3 | ↑ 10.3 |
| + JavaScript | 9.4 | ↑ 8.4 |
| + Python (SWE-Mirror) | 10.8 | ↑ 9.8 |
| + Python (SWE-rebench) | 11.4 | ↑ 10.4 |

Table 4: Performance on SWE-Bench-Verified of models trained on 512 trajectories each language from SWE-Mirror-60K and 512 Python trajectories from SWE-rebench(Real).

with actions for operating computer like editors and shells. In contrast to relying on an LLM's autonomous decision-making, another line of research argues for utilizing structured workflow architectures. Agentless (Xia et al., 2024), Agentless-Mini (Wei et al., 2025) and Moatless (moa) demonstrate that combining workflow with test-time scaling can outperform many sophisticated SWE agents on SWE-bench while reducing computational costs. Some research works also explored the self-evolution of coding agents, exemplified by GDM (Zhang et al., 2025), SE-Agent (Lin et al., 2025) and SWE-Exp (Chen et al., 2025), showing impressive improvement. Another research area has focused on enhancing the models themselves. SWE-Fixer (Xie et al., 2025) represents a learning-based approach, improving file retrieval and patch generation capabilities using supervised fine-tuning to effectively train open-source LLMs for specialized SWE tasks. SWE-Gym (Pan et al., 2025) and SWE-Smith (Yang et al., 2025b) have explored rejection sampling fine-tuning, an approach that we also adopt in our work. Furthermore, reinforcement learning (RL) has been utilized to refine model capabilities, with SWE-RL (Wei et al., 2025) using patch similarity as a reward signal and SWE-Swiss (swe), DeepSWE (AI, 2025) and SkyRL (Cao et al., 2025) exploring execution-based rewards as a promising future direction.

**Issue-Resolving Datasets.** The development of datasets for training and evaluating issue-resolving agents has rapidly progressed from static code collections to dynamic, interactive environments. A foundational contribution is SWE-Gym (Pan et al., 2025), which established the paradigm of using real-world Python issues paired with executable environments and unit tests, enabling interactive agent training and verification. To combat the growing problem of data contamination in static benchmarks, SWE-rebench (Badertdinov et al., 2025) and SWE-Factory (Guo et al., 2025) introduced a dynamic pipeline that continuously sources fresh, decontaminated tasks from active GitHub repositories, ensuring a more robust and reliable evaluation of an agent's true generalization capabilities. Recognizing that manual curation remains a significant bottleneck, subsequent efforts have focused on scalable, automated data generation. SWE-Smith (Yang et al., 2025b) pioneered a synthetic approach by inverting the typical workflow, starting with working code and automatically injecting bugs to create thousands of new tasks. Similarly, SWE-Synth (Pham et al., 2025) uses LLMs to simulate the entire debugging process, generating not just code fixes but also test cases and structured repair trajectories. Complementing these, R2E-Gym (Jain et al., 2025) leverages a procedural generation pipeline to curate large-scale training environments directly from code commits, reducing the reliance on human-written issues. Together, these works highlight a critical trend towards creating more scalable, realistic, and verifiable data sources to advance agentic coding.

## 5 CONLUSION

This paper introduces SWE-Mirror, a novel pipeline which multiplies the utility of each Gym and unlocks the vast history of software evolution on platforms like GitHub as a source of training data. Our primary contribution is the release of SWE-Mirror-60K, a large-scale dataset of 60,000 verifiable tasks built using this methodology. Our empirical evaluations demonstrated that models finetuned on SWE-Mirror-60K exhibit significant improvements in their issue-resolving capabilities, validating the quality and effectiveness of our approach. Furthermore, our in-depth ablation studies provide critical insights for the field. We have also confirmed a strong scaling law where performance consistently improves with data volume, demonstrated the efficiency of *Error Masking* training strategy and revealed the evidence of cross-lingual generalizability, where models trained exclusively on non-Python data still exhibit notable proficiency on Python tasks, highlighting the value of multi-lingual data in learning generalized, abstract problem-solving patterns.

ETHICS STATEMENT

This work adheres to the ICLR Code of Ethics. All datasets used were sourced in compliance with relevant usage guidelines, ensuring no violation of privacy. We have taken care to avoid any biases or discriminatory outcomes in our research process. No personally identifiable information was used, and no experiments were conducted that could raise privacy or security concerns. We are committed to maintaining transparency and integrity throughout the research process. All data from the evaluation benchmarks are not included and strictly filtered in any phase of SWE-MIRROR to avoid contamination.

REPRODUCIBILITY STATEMENT

We have made every effort to ensure that the results presented in this paper are reproducible. All code have packaged in supplementary mateirials to facilitate replication and verification. The experimental setup, including training steps, model configurations, and hardware details, is described in detail in Section 3. All code, datasets and models will be open-sourced. We believe these measures will enable other researchers to reproduce our work and further advance the field.

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

# A APPENDIX

## A.1 PULL-REQUEST COLLECTION AND FILTER

We use following rules to collect high-quality pull-requests:

- It must have linked issues;

- It must have been merged and closed;

- It must edit code files.

Unlike SWE-bench (Jimenez et al., 2024), our filtering criteria do not require pull requests to modify test files. This is for two reasons: first, it is difficult to isolate test modifications in languages like Rust where tests are co-located with source code; second, we generate tests separately using a *Test Agent*. To finalize our dataset, we use an LLM for quality control and to predict mirrorability, as guided by the following prompt.

---

**Prompt for LLM Filter**

```
prompt = """You are a senior software engineer.

You are given a pull request from another repository.

You are going to check, response True in final answer if the pull request is a bug
    ↪ fix or a feature addition, and response False if the pull request is just
    ↪ fixing some error messages or documentations.
1. is the pull request a bug fix or a feature addition
2. is the pull request non-trivial, just fixing error messages, docs, also, this
    ↪ not-related to external dependencies.
3. if some functionality related to the bug or feature exists in the current
    ↪ repository.

Belowing is the description of the pull request:
<pull_request>
    <body>
    {body}
    </body>
    <diff>
    {diff}
    </diff>
</pull_request>

Belowing is the readme and the test suite of the current repository:
<current_repo>
    <readme>
    {readme}
    </readme>
    <test_suite>
    {test_suite}
    </test_suite>
</current_repo>

Think Step by Step with following questions
1. What is the bug fixed or the feature added in the pull request?
2. What is the related functionality of the bug?
3. Does the current repository have the related functionality:
    1. If yes, what is the related functionality?
4. Is it possible to introduce the bug/feature in the current repository?

Note:
- The language of repos does not matter, you should focus on the functionality of
    ↪ the bug.

Respond with python list with two elements, "exists", "reason", in the following
    ↪ format:
```python
[True/False, "The pull request is a bug fix or a feature addition, related to
    ↪ ...., the current repository has the related functionality."]
```
"""
```

---

## A.2 TASK MIRRORING WORKFLOW

The first step is to distill the core symptoms and logic from pull requests in similar repositories. We introduced this step for a critical reason: raw pull request and issue descriptions often contain repository-specific information (*e.g.*, variable names, file paths, and stack traces). This context-specific data can mislead the model into localizing non-existent files or generating patches that result in compilation or syntax errors. The distillation process, therefore, focuses on extracting the underlying functionality, core logic, current and expected behavior, and observable symptoms. The prompts used for this task are provided below.

---

**Prompt for Problem Abstraction**

```
Consider the following pull request that fixes a bug:
<pull_request>
    <body>
    {body}
    </body>
    <diff>
    {diff}
    </diff>
</pull_request>

Your task is to abstract the bug pattern from the pull request, focusing
    ↪ exclusively on systemic issues that require changes in multiple locations
    ↪ across the codebase.

Here is an example of a complex bug pattern that requires multiple edits:
<pull_request>
    <body>
    Fix inconsistent error handling across API endpoints

    Multiple API endpoints were handling validation errors differently, leading to
        ↪ inconsistent error responses and poor user experience. Some endpoints
        ↪ returned 400 status codes while others returned 500, and error message
        ↪ formats varied. This PR standardizes error handling across all
        ↪ user-facing endpoints to provide consistent behavior.

    The fix involves:
    - Updating user registration endpoint error handling
    - Fixing profile update validation responses
    - Standardizing login error messages
    - Adding consistent error formatting in shared utilities

    Fixes #456
    </body>
    <diff>
    @@ -8,7 +8,8 @@ class UserController:
        def register(self, user_data):
            if not self.validate_user_data(user_data):
    -           return {{"error": "Bad input"}}, 500
    +           return {{"error": "Invalid user data", "details":
        ↪ self.get_validation_errors(user_data)}}, 400

    @@ -22,7 +23,8 @@ class UserController:
        def update_profile(self, user_id, profile_data):
            if not self.validate_profile_data(profile_data):
    -           raise Exception("Validation failed")
    +           return {{"error": "Invalid profile data", "details":
        ↪ self.get_validation_errors(profile_data)}}, 400

    @@ -35,6 +37,7 @@ class AuthController:
        def login(self, credentials):
            if not self.validate_credentials(credentials):
    -           return {{"message": "Login failed"}}, 500
    +           return {{"error": "Invalid credentials", "details": "Username or
        ↪ password incorrect"}}, 401

    @@ -5,6 +5,10 @@ class ValidationUtils:
    +   def get_validation_errors(self, data):
    +       # Standardized error formatting
    +       return [str(error) for error in self.validator.errors(data)]
    +
        def validate_user_data(self, data):
            return self.validator.is_valid(data)
    </diff>
```

---

```
</pull_request>

Follow this pattern when abstracting the bug - identify systemic issues that
    ↪ manifest across multiple files and functions:

```md
### Bug Pattern

**Issue Type**: Inconsistent Error Handling / API Response Standardization

**Core Problem**:
The application lacks consistent error handling patterns across similar functions
    ↪ or modules, leading to unpredictable behavior and poor user experience.
    ↪ Different parts of the codebase handle similar error conditions in
    ↪ incompatible ways.

**Technical Context**:
- API endpoints or service methods that perform similar validation or processing
- Error handling logic scattered across multiple controllers, services, or utility
    ↪ functions
- Inconsistent status codes, error message formats, or exception handling
    ↪ approaches
- Missing standardized error response structures

**Symptom**:
- Different error responses for similar failure conditions
- Inconsistent HTTP status codes across related endpoints
- Varying error message formats that confuse API consumers
- Some functions throw exceptions while others return error objects

**Root Cause Pattern**:
- Lack of centralized error handling utilities or standards
- Copy-paste development without following established patterns
- Missing shared validation or error formatting functions
- Inconsistent exception handling strategies

**Impact Scope**:
Multiple locations typically affected:
- All API endpoints that perform user input validation
- Service layer methods that process similar data types
- Controller functions handling authentication or authorization
- Utility functions used for data processing or validation
- Error response formatting across different modules
```

Please wrap the bug pattern in the following format:
```md
.. the bug pattern ..
```
"""
```

For *Test Agent* and *Mirror Agent*, we implement them in Agentless style, each go through: (1) localize related file and (2) genearte patch in *Search/Replace* format.

**Test Agent: Prompt for Localization**

```
TEST_LOCALIZE = """\
Please look through a given issue description and repository structure and provide
    ↪ two list of files related to the issue:
- `source_files`: the files may contains code related to the functionality
    ↪ described in the issue
- `test_files`: the files which should contain the test cases for the
    ↪ functionality described in the issue

--- BEGIN ISSUE ---
{issue}
--- END ISSUE ---

--- BEGIN REPOSITORY STRUCTURE ---
{structure}
--- END REPOSITORY STRUCTURE ---

Only provide the full path and return at most {n} files for each list.
```

```
Respond in the following format, wrapped your results in a markdown python code
    ↪ block with a dictionary with two keys `source_files` and `test_files`.
```python
{{
    "source_files": [
        "most/important/file1.xx",
        "less/important/file2.yy",
        ...
    ],
    "test_files": [
        "most/important/file1.xx",
        "less/important/file2.yy",
        ...
    ]
}}
```
```

**Test Agent: Prompt for Patch Generation**

```
TEST_PATCHGEN = """We are currently adding unit tests to the avoid the future
    ↪ regression for functionality described in the issue.

--- BEGIN ISSUE ---
{issue}
--- END ISSUE ---

Below are some source code segments related to the functionality described in the
    ↪ issue.

--- BEGIN SOURCE FILES ---
{source_files}
--- END SOURCE FILES ---

Below are some files you can edit to add unit tests.
--- BEGIN TEST FILES ---
{test_files}
--- END TEST FILES ---

Please first localize the code in SOURCE FILES to the functionality described in
    ↪ the issue and \
then generate *SEARCH/REPLACE* edits to test to some of TEST FILES to test the
    ↪ issue.

Every *SEARCH/REPLACE* edit must use this format:
1. The file path
2. The start of search block: <<<<<<< SEARCH
3. A contiguous chunk of lines to search for in the existing source code
4. The dividing line: =======
5. The lines to replace into the source code
6. The end of the replace block: >>>>>>> REPLACE

Here is an example:

```
{diff_example}
```

Please note that the *SEARCH/REPLACE* edit REQUIRES PROPER INDENTATION. If you
    ↪ would like to add the line '        print(x)', you must fully write that
    ↪ out, with all those spaces before the code!
Wrap each *SEARCH/REPLACE* edit in a code block as shown in the example above. If
    ↪ you have multiple *SEARCH/REPLACE* edits, use a separate code block for
    ↪ each one.

Please make sure the tests you add are not too simple and can be passed by the
    ↪ existing code.
"""
```

**Mirror Agent: Prompt for Localization**

```
MIRROR_LOCALIZE = """\
Please look through a given issue description, repository structure, a patch
    ↪ related to test the issue and provide a list of files related to the issue
```

```
        Below is the issue description and repository structure.
        --- BEGIN ISSUE ---
        {issue}
        --- END ISSUE ---

        Below is the repository structure.
        --- BEGIN REPOSITORY STRUCTURE ---
        {structure}
        --- END REPOSITORY STRUCTURE ---

        Below is the patch applied to the repository to test the issue.
        --- BEGIN TEST PATCH ---
        {testgen_patch}
        --- END TEST PATCH ---

        Only provide the full path and return at most {n} files.

        Respond in the following format, wrapped your results in a markdown python code
            ↪ block with a list of files.
        ```python
        [
            "most/important/file1.xx",
            "less/important/file2.yy",
            ...
        ]
        ```

        """.strip()
```

### Mirror Agent: Prompt for Patch Generation

```
        MIRROR_PATCHGEN = """We are currently implementing the issue described in the
            ↪ following issue description.

        --- BEGIN ISSUE ---
        {issue}
        --- END ISSUE ---

        Below are some code segments related to the issue.

        --- BEGIN FILES---
        {files}
        --- END FILES---

        Below is the patch applied to the repository to test the issue, please DO NOT
            ↪ modify any test code or test files.
        --- BEGIN TEST PATCH ---
        {testgen_patch}
        --- END TEST PATCH ---

        Here is the list of testcases related to the issue.
        --- BEGIN TESTS ---
        {tests}
        --- END TESTS ---

        Please first localize the related source code based on the issue description, and
            ↪ then generate *SEARCH/REPLACE* edits to re-implement the issue via breaking
            ↪ the tests in the TESTS section.
        DO NOT modify any test code or test files, you should only modify the non-test
            ↪ files and code related to the issue.

        Every *SEARCH/REPLACE* edit must use this format:
        1. The file path
        2. The start of search block: <<<<<<< SEARCH
        3. A contiguous chunk of lines to search for in the existing source code
        4. The dividing line: =======
        5. The lines to replace into the source code
        6. The end of the replace block: >>>>>>> REPLACE

        Here is an example:
```

```
```
{diff_example}
```

Please note that the *SEARCH/REPLACE* edit REQUIRES PROPER INDENTATION. If you
    ↪ would like to add the line '        print(x)', you must fully write that
    ↪ out, with all those spaces before the code!
Wrap each *SEARCH/REPLACE* edit in a code block as shown in the example above. If
    ↪ you have multiple *SEARCH/REPLACE* edits, use a separate code block for
    ↪ each one.
"""
```

## A.3 FRAMEWORK ANALYSIS

To assess the effectiveness and fidelity of SWE-MIRROR, we conducted a detailed analysis of our framework. Our goal was to answer three core questions: (1) How effective is our LM-based pre-filter? (2) What is the end-to-end mirror success rate for promising candidates? (3) Are the final mirrored tasks semantically consistent with the original issues and seems realistic?

**Effectiveness of LM-based Pre-filter.** A critical component of our framework's efficiency is the LM-based heuristic, which acts as an intelligent filter to identify *high-quality* and *mirrorable* tasks in Section 2.1. To rigorously evaluate its performance, we constructed a balanced evaluation set of 100 issues manually select from issues after the rule-based filtering. This set contains 50 positive instances, which are high-quality and mirrorable, and 50 negative instances, comprising issues that are either low-quality or impossible to mirror. The filter's task is to accept the positive instances while rejecting the negative ones. As Table 5 shown, the filter demonstrates a high precision of 84.3%. This ensures that the vast majority of issues passed to the expensive downstream stages are indeed valuable candidates, thus minimizing wasted computation. Furthermore, with a recall of 86.0% , the filter successfully captures a large portion of the usable issues.

|          | Accepted | Rejected |
|----------|----------|----------|
| Positive | 43       | 7        |
| Negative | 8        | 42       |

Table 5: Confusion matrix for the LM-based filter.

| Language   | Yield Rate (%) | Error(%)       |          |
|------------|----------------|----------------|----------|
|            |                | Compile/Syntax | Semantic |
| Python     | 68.0           | 2.0            | 30.0     |
| Rust       | 28.0           | 36.0           | 36.0     |
| Go         | 36.0           | 28.0           | 36.0     |
| JavaScript | 52.0           | 6.0            | 42.0     |
| **Overall**    | **46.0**           | **18.0**           | **36.0**     |

Table 6: Detailed breakdown of outcomes from the task mirroring phase, with error types categorized.

**Effectiveness of Mirroring.** We next evaluate the core of our framework: the task mirroring engine. The goal here is to measure the success rate when the pipeline is provided with ideal inputs. For this experiment, we manually select each 100 issues for Python, Rust, Go and Javascript following the same criteria as previous experiment. Result is considered success if it passed the validation in Section 2.3. To gain deeper insight into the failure modes, we further categorized each unsuccessful attempt into one of two types. The first is *Compile/Syntax Error*, which we define as any instance where no tests could be run, typically because the generated patch prevents the project from building or leads to a fatal syntax error. The second is *Semantic Error*, which encompasses all other failures

where the code runs, but does not correctly produce the required "fail-to-pass". The results, presented in Table 6, show an overall yield rate of **46.0%**. Performance, however, varies significantly by language. Python achieves the highest success rate at 68.0%, while compiled languages like Rust (28.0%) and Go (36.0%) prove more challenging. The error breakdown reveals why: *Compile/Syntax* errors are the dominant failure mode for Rust and Go, accounting for 36.0% and 28.0% of their respective totals. In contrast, this error type is rare for the dynamically-typed Python (2.0%) and JavaScript (6.0%).

Table 7: The human classify results on the semantic faithfulness of 184 mirrored tasks.

| | **Agreement Pattern** | | |
|---|---|---|---|
| **Final Classification** | **Unanimous (3-0)** | **Majority (2-1)** | **Total** |
| High Consistency | 90 | 25 | 115 |
| Moderate Consistency | 30 | 11 | 41 |
| Inconsistent | 15 | 6 | 21 |
| Unclassifiable (No Majority) | | | 7 |

**Faithfulness of Mirroring.** A high yield rate is only meaningful if the generated tasks are faithful representations of the original problems. A task that passes our validation but does not reflect the source issue's core logic is not a useful addition to a dataset. Therefore, our final analysis evaluates the semantic fidelity of the successfully mirrored tasks. To assess this, the 184 tasks successfully generated in Section A.3 were independently audited by three human annotators. They compared each generated task instance against the original GitHub issue and PR pair. The results of this audit were highly encouraging. As shown in Table 7, a consensus was reached on the vast majority of tasks. Out of the 177 tasks with majority results, 156 tasks (88.1%) were deemed to have either High or Moderate consistency, providing strong evidence that SWE-MIRROR succeeds in preserving the semantic essence of real-world software engineering challenges.

A.4 DIVERSITY ANALYSIS OF SYNTHESIZED ISSUES

To evaluate whether SWE-Mirror merely replicates existing patterns or genuinely expands the scope of problem scenarios, we conducted a quantitative diversity analysis. Since issue-type labels are not standardized across repositories and require subjective manual annotation, we adopted **Semantic Cluster Entropy** as an objective, embedding-based metric to measure diversity directly from the problem statement text.

**Methodology.** We focused our analysis on the `dask/dask` repository, which is well-represented in both our generated dataset (SWE-Mirror) and the real-world baseline (SWE-rebench). The procedure was as follows:

A.5 DIVERSITY ANALYSIS

To verify that SWE-Mirror expands problem diversity beyond historical patterns, we computed **Semantic Cluster Entropy** on the `dask/dask` repository. We randomly sampled 100 problem statements each from SWE-Mirror and the real-world baseline (SWE-rebench), embedded them using `text-embedding-3-large`, and performed K-means clustering ($k = 10$) on the combined embedding space. SWE-Mirror achieved a significantly higher cluster entropy of **3.21** compared to **1.55** for real data (a $2.07\times$ increase). This indicates that our approach covers a broader and more uniformly distributed semantic space, successfully introducing diverse logic from external sources into the target environment.

B UTILIZATION OF LARGE LANGUAGE MODELS

In the development of this research, large language models (LLMs) were utilized to refine the manuscript, conduct thorough literature reviews, and generate visualizations.

