# OpenReview forum: "SWE-Mirror: Scaling Issue Resolving Datasets by Mirroring Issues Across Repositories"
_ICLR.cc/2026/Conference — Submitted to ICLR 2026_

### Official Review · Reviewer_wqTS · 2025-10-29

**Soundness:** 3
**Presentation:** 3
**Contribution:** 3
**Rating:** 6
**Confidence:** 4

**Summary:**

This paper addresses the challenge of constructing large-scale, verifiable datasets for issue-resolving tasks. The authors propose SWE-MIRROR, a pipeline that mirrors real-world issues into repositories with pre-configured Gym environments, enabling verifiable task synthesis without costly environment setup.

**Strengths:**

- The motivation is sound — SWE-MIRROR is proposed to address the limitations of existing SWE training data construction methods.
- Experimental results show that models trained with SWE-MIRROR outperform the baselines.

**Weaknesses:**

- The repository diversity is relatively low, which may pose risks to the model’s generalization ability.
- The paper lacks an analysis of the diversity of synthesized issues. For example, are the synthesized issues more diverse in terms of issue types compared to those within a single real repository? Such an analysis would help better understand the effectiveness of SWE-MIRROR.
- In Table 4, the ablation study on training data is conducted only on a subset of 512 trajectories; it remains unknown whether the gap between synthesized and real data would widen as the dataset scales up.

**Questions:**

- When constructing a large amount of data for a single repository, would the LLM overfit or memorize that repository’s content, thereby harming generalization?
- In the Task-Mirroring stage, why is the test generated before the patch, rather than generating the patch first and then the test?
- In line 295, why do you “keep the model’s only the last 5 observations’ content from environment in the context”?

---

> ### Author Response · Authors · 2025-11-28
> **Author Response [1/2]**
>
> Thank you for the thoughtful review. Below, we address your comments point-by-point and outline our revisions.
>
> > `Weakness1 & Question1` The repository diversity is relatively low, which may pose risks to the model’s generalization ability.... When constructing a large amount of data for a single repository, would the LLM overfit or memorize that repository’s content, thereby harming generalization?
>
> ---
>
> **Answer1**: We appreciate this insightful question. To address your concern, we structure our response into two parts:
>
> - **SWE-Mirror does not harm generalization**:  Our empirical results confirm that training on a massive volume of tasks within a limited set of repositories does not lead to overfitting or memorization. Instead, it fosters robust capabilities that transfer effectively:
>     - *Cross-repository generalization*: Our model achieves strong performance on SWE-Bench-Verified and Multi-SWE-Bench, whose repositories are entirely disjoint from our training sources. This indicates that the **model learns repository-agnostic repair patterns** rather than merely memorizing project-specific content. This finding also align with previous work like SWE-Smith[1], in which paper authors found that training massiviely only on one repository can also increase model's behaviour on SWE-bench's other repository.
>     - *Cross-lingual generalization*: Our study on cross-lingual generalization in Section 3.3.2 further shows that the model captures transferable reasoning patterns that extend beyond specific repositories and even across programming languages.
> - **Rationale for the current limited 40 repositories**: we select environment from open-source datasets in SWE-rebench and Multi-SWE-RL, after filtering(time limit < 5 min, memory limit < 1GB) and manual inspect the correctness of environment setup, we only get the 40 environments as the startpoint. We plan to expand diversity in future iterations by integrating automated environment setup tools and relaxing resource constraints and increasing the machines we use for docker containers running.
>
> > `Weakness2`: The paper lacks an analysis of the diversity of synthesized issues. For example, are the synthesized issues more diverse in terms of issue types compared to those within a single real repository? Such an analysis would help better understand the effectiveness of SWE-Mirror.
>
> **Answer2**: Thank you for highlighting the importance of measuring the diversity of synthesized issues. We have updated the paper about how SWE-Mirror expands the diversity of issues compared to those within a single real repository in Appendix A.4.
>
> Since issue-type labels are not standardized and require manual categorization and annotation, we instead quantify diversity directly from the problem-statement text using semantic cluster entropy, a standard embedding-based diversity metric. We use repository dask/dask for this analysis, which appears in both SWE-Mirror and SWE-rebench. We randomly sample 100 problem statements of dask/dask from each dataset, embed them using the text-embedding-3-large model, and perform K-means clustering over the combined embedding set. For each dataset, we compute the entropy of its cluster-membership distribution.
>
> SWE-Mirror achieves higher cluster entropy than the real issues in SWE-rebench (1.55 vs. 3.21, a **2.07×** increase). The results demonstrate that SWE-Mirror not only increases issue quantity but also substantially expands diversity within the same repository.
>
> below is the code for computing semantic cluster entropy.
> ```
> def compute_cluster_entropy_once(
>     real,
>     mirror,
>     sample_size=100,
>     n_clusters=10,
>     embedding_model="openai/text-embedding-3-large"
> ):
>
>     def embed_texts(texts):
>         embs = []
>         for t in texts:
>             res = embedding(model=embedding_model, input=t)
>             embs.append(res["data"][0]["embedding"])
>         return np.array(embs)
>
>     def cluster_entropy(cluster_ids, k):
>         cnt = Counter(cluster_ids)
>         total = len(cluster_ids)
>         entropy = 0.0
>         for c in range(k):
>             p = cnt[c] / total if c in cnt else 0
>             if p > 0:
>                 entropy -= p * math.log(p + 1e-12)
>         return entropy
>
>     all_texts = real + mirror
>     emb_all = embed_texts(all_texts)
>     kmeans = KMeans(n_clusters=n_clusters, random_state=0).fit(emb_all)
>     cluster_ids = kmeans.labels_
>     real_clusters = cluster_ids[:sample_size]
>     mirror_clusters = cluster_ids[sample_size:]
>     entropy_real = cluster_entropy(real_clusters, n_clusters)
>     entropy_mirror = cluster_entropy(mirror_clusters, n_clusters)
>
>     return {
>         "entropy_real": entropy_real,
>         "entropy_mirror": entropy_mirror,
>     }
> # results = compute_cluster_entropy_once(real_issues, mirror_issues)
> # print(results)
> ```

---

> ### Author Response · Authors · 2025-11-28
> **Author Response [2/2]**
>
> > `Weakness3` In Table 4, the ablation study on training data is conducted only on a subset of 512 trajectories; it remains unknown whether the gap between synthesized and real data would widen as the dataset scales up.
>
> **Answer3**: Synthesized data inevitably contains noise or lacks the subtle nuances of human-curated solutions[3]. However, we argue that the primary advantage of synthetic data lies in its scalability and diversity, which compensate for the gap in individual data quality.
>
> Also It is important to note that the current filtering process of SWE-Mirror only ensures the synthetic data is **correct** (i.e., using test status under three repository state), without applying the additional quality assessments used in the compared baseline, SWE-rebench [2]. This suggests that our data quality has significant room for further improvement through stricter filtering.
>
>
> To better clarify our contribution. SWE-Mirror offers key advantages that real data cannot easily achieve:
> - **Scalability** – thousands of problem statements can be generated per repository without human effort.
> - **Semantic diversity** – as shown in our cluster-entropy analysis, SWE-Mirror covers a broader and more varied space of issue semantics.
> - **Practical feasibility** – collecting equivalent amounts of high-quality real data is extremely costly; for example, SWE-Bench requires several hours per issue for labeling and verification.
> Together, these factors demonstrate that SWE-Mirror is a practical, effective, and complementary resource to real data, enabling experiments at scales that would be infeasible with manually curated datasets.
>
> > `Question2` In the Task-Mirroring stage, why is the test generated before the patch, rather than generating the patch first and then the test?
>
> **Answer4**: We generate the test before the patch for two main reasons:
> - **Able to early exit and cost efficiency**: Generating the test first allows us to verify the test before introducing the bug. If the generated test fails, we can exit early, avoiding unnecessary patch generation and additional LLM queries.
> - **Avoid overfitting on bugs**: If the patch is generated first, then generating a test against that patch risks the test overfitting to the bug or even to the specific patch, reducing its generalization value. By generating the test first, we ensure that it captures the intended behavior independently of the patch.
> Question3: In line 295, why do you “keep the model’s only the last 5 observations’ content from environment in the context”?
> Reasons are described in the **Evaluation Benchmarks and Metrics** paragraph in Section 3.1. We limit the 7B model’s training and inference to a 32K context length to avoid excessive computation in ablation study. Additionally, we keep only the last 5 observations, which both prevents context-length errors and maintains consistency with prior work (SWE-Smith), ensuring fair comparison and stable performance.
>
> ---
>
> [1] SWE-smith: Scaling Data for Software Engineering Agents
>
> [2] R2E-Gym: Procedural Environments and Hybrid Verifiers for Scaling Open-Weights SWE Agents
>
> [3] Magicoder: Empowering Code Generation with OSS-Instruct
>
> ---

---

### Official Review · Reviewer_hCtU · 2025-10-31

**Soundness:** 3
**Presentation:** 3
**Contribution:** 2
**Rating:** 4
**Confidence:** 4

**Summary:**

The paper studies the problem of scaling training data for real-world SWE tasks. Manual curation and filtering of SWE environments using human PRs is inefficient and not scalable. The paper thus proposes a synthetic approach which mirrors issues in other repositories in existing gym-environments. While the approach for synthetic curation is similar to some prior works, the results and cross-lingual generalization is interesting.

**Strengths:**

* The results for cross-lingual generalization is interesting.

* The paper shows the power of synthetic data for SWE training. While their are prior works like SWE-Smith and R2E-Gym which also use synthetic data for SWE training, the proposed approach is interesting.

* The final performance on SWE-Bench Verified is impressive.

**Weaknesses:**

* Table 2 lacks comparison with recent methods on also scaling SWE tasks [1]. While the authors mention it in the end in related work, I feel its important to provide a discussion early on on the comparison with other methods for scaling SWE tasks.
    * This seems especially relevant in context to recent works like SWE-Smith and R2E-Gym which have similar motivation as the paper (L51) i.e. "scaling SWE tasks manually or through human written PRs alone is not scalable".
    * Similarly, in Sec. 3.4 when the authors evaluate the impact o f synthetic data on SWE training mentioning "Synthetic data quality is competitive with real data", it will be useful to also analyze how the proposed synthetic data compares with prior synthetic data generation methods for SWE tasks.

* As mentioned in L182-188, the "A different agent-Mirror
Agent takes the abstract description for semantic context and the file paths and function names from the test.patch as a strong structural prior. Its objective is to surgically modify the application’s source code to specifically cause the new test case to fail". This seems very similar conceptually to SWE-Smith which also mnodifies the existing codebases to cause new bugs to appear.

* Similarly the "Mirroring Problem Statement" in L189-196 seems very similar to the backtranslation approach in SWE-Smith and also in some part with R2E-Gym which also generates new problem statements given ground truth patch, relevant files and context etc.

* Finally, task verification is also similar to other works like SWE-Gym and SWE-Smith.

* Is the 52.2 reported in Table 3, pass@1 or Best@K with test-time scaling?

* Also another concern for Table 3 will be that prior works use worse teacher models for RFT on SWE trajectories. Thus, it will be important to analyze the performance improvements under the same teacher model. This might help to disentangle and emphasize the role of data quantity for SWE training - which is the main contribution of the paper.

References:
[1] R2E-Gym: Procedural Environments and Hybrid Verifiers for Scaling Open-Weights SWE Agents, 2025

[2] SWE-smith: Scaling Data for Software Engineering Agents, 2025

**Questions:**

Please see the weaknesses section for some additional questions.

---

> ### Author Response · Authors · 2025-11-28
> **Author Response [1/3]**
>
> Thank you for the comments and. We appreciate the opportunity to clarify our contributions and present new experimental results that directly address the concerns regarding baselines and data quality.
>
> > Weakness1& Weakness1.1: Table2 lacks comparison with recent methods on also scaling SWE tasks [1]. While the authors mention it in the end in related work, I feel its important to provide a discussion early on on the comparison with other methods for scaling SWE tasks... This seems especially relevant in context to recent works like SWE-Smith and R2E-Gym which have similar motivation as the paper (L51) i.e. "scaling SWE tasks manually or through human written PRs alone is not scalable".
>
> **Answer1**: We have included the discussion of those works in the introduction (Lines 58-64).
> While we share the ultimate goal of scaling SWE tasks and the premise that manual curation is insufficient, SWE-Mirror is grounded in distinct observations and motivations, leading to a fundamentally different methodology:
> - R2E-Gym[1] and SWE-Smith[2] (Internal Exploitation): These motivation and methods primarily focus on mining internal resources. They rely on "back-translation" to generate issues from existing commits (R2E-Gym) or use mutation-based synthesis  and past pull-requests (SWE-Smith) within a specific repository.
> - SWE-Mirror (External Exploration): We aim to bridge the gap between scarce functional environments and the abundant "ocean" of external open-source PRs. Our core observation is the "mirroring issue" phenomenon: while executable environments are hard to build, valid coding logic is ubiquitous. SWE-Mirror abstracts "Portable Problem Logic" from external, unrelated PRs and transplants this semantic essence into verified Gym environments. This allows us to tap into a vast, diverse source of real-world logic that previous methods ignore.
>
> > `Weakness1.2 & Weakness6`: Similarly, in Sec. 3.4 when the authors evaluate the impact of synthetic data on SWE training mentioning "Synthetic data quality is competitive with real data", it will be useful to also analyze how the proposed synthetic data compares with prior synthetic data generation methods for SWE tasks... Also another concern for Table 3 will be that prior works use worse teacher models for RFT on SWE trajectories. Thus, it will be important to analyze the performance improvements under the same teacher model. This might help to disentangle and emphasize the role of data quantity for SWE training - which is the main contribution of the paper.
>
> **Answer2**: Thank you for pointing out that prior works often use weaker teacher models, which makes it difficult to disentangle the impact of data quantity versus quality. To address this, we conducted a strictly controlled ablation study.
> - Settings: We compared SWE-Mirror against baselines using the same backbone model, same teacher model, same scaffold, and same amount of training trajectories.
>   - Backbone Model: Qwen2.5-Coder-Instruct-7B
>   - Teacher Model: Claude-4-Sonnet
>   - Scaffold: OpenHands
>   - Trajectories: 512
>
> | **Dataset** | **Type** | **SWE-Bench-Verified** | **Tasks per Gym** | **Tasks per Repo** |
> | :--- | :--- | :---: | :---: | :---: |
> | **SWE-rebench** | Real | 11.4% | 1 | 6.15 |
> | **SWE-Smith** | Synthetic | 10.2% | 266.38 | 266.38 |
> | **R2E-Gym** | Synthetic | 10.6% | 1 | 623.15 |
> | **SWE-Mirror (Ours)** | **Synthetic** | **10.8%** | **1,516.78** | **1,516.78** |
>
> The results lead to two critical strength of SWE-Mirror:
> 1. **Comparable Quality**: Under identical settings, SWE-Mirror achieves a resolve rate (10.8%) comparable to real-world data (11.4%), validating that our "mirrored" tasks provide a training signal as high-quality as authentic human-curated issues.
> 2. **Superior Scalability**: Since the quality is established, the decisive factor becomes scale. As demonstrated in our Scaling Law analysis (Section 3.3.1), performance improves consistently with data volume. Unlike real-world data which is finite and hard to acquire, SWE-Mirror allows us to generate *>5x* more tasks per Gym/Repo than prior synthetic methods (see Table above). By combining "real-world quality" with "synthetic scalability," we can push further along the scaling curve to achieve state-of-the-art performance.

---

> ### Author Response · Authors · 2025-11-28
> **Author Response [2/2]**
>
> > *Weakness2 & Weakness3 & Weakness4** As mentioned in L182-188, the "A different agent-Mirror Agent takes the abstract description for semantic context and the file paths and function names from the test.patch as a strong structural prior. Its objective is to surgically modify the application’s source code to specifically cause the new test case to fail". This seems very similar conceptually to SWE-Smith which also modifies the existing codebases to cause new bugs to appear. ... Similarly the "Mirroring Problem Statement" in L189-196 seems very similar to the backtranslation approach in R2E-Gym and also in some part with SWE-Smith which also generates new problem statements given ground truth patch, relevant files and context etc. .... Finally, task verification is also similar to other works like SWE-Gym[4] and SWE-Smith.
>
> **Answer3**: Thank you for noting the **high-level conceptual** structural similarities . We believe these similarities stem from the **intuitive nature** of making a issue-resolving datasets: finding(SWE-Gym/R2E-Gym) or introduce(SWE-Smith/SWE-Mirror) an issue (and subsequently verifying them) is the most natural and logical paradigm for creating a issue-resolving task. To synthesize a verifiable issue-resolving task, one must inevitably establish a "buggy" state. *Introducing a new issue without modifying the codebase is logically impossible.*
>
> However, while the high-level framework is shared, SWE-Mirror differs fundamentally in methodology and information sources.
>
> 1. **Issue Introudction**
> - SWE-Smith: Primarily relies on random/LLM-based mutations or mechanically reverting PRs within the same repository. While this breaks tests, the resulting defects often lack semantic depth or represent "noise" rather than logical bugs.
> - SWE-Mirror: We utilize significantly richer information by employing the original Issue as a strong semantic prior. Our Mirror Agent does not just randomly mutate code; it uses the real-world issue to "surgically" inject a bug that is logically consistent with the original real-world issue's intent.
>
> 2. **Problem Statement Generation**
> - R2E-Gym/SWE-Smith(Summarization):  typically generates problem statements by summarizing the code diff or context directly.
> - SWE-Mirror (Reconstruction): Our approach is not just summarizing the code; it is a semantic reconstruction. Because our bug injection is grounded in the original issue, our "Mirroring" step reconstructs a problem statement that aligns with the specific misunderstanding introduced in the bug. This ensures the synthetic <Issue, Bug> pair mimics the complex causal reasoning found in real-world maintenance tasks, rather than simple code-to-text translation.
>
> 3. **Verification**:
> - Regarding task verification, all these works adhere to the **standard established by SWE-Bench**, the standard definition of correctness for this field, not a methodological design choice that allows for variation.
> - Furthermore, we enhanced our evaluation harness to ensure greater robustness during both task verification and trajectory collection. Instead of validating only the tests modified by the patch (i.e., test.patch), we enforce the execution of the entire repository test suite. As highlighted in recent research[4], relying on partial testing can miss regression bugs, leading to a false positive rate of approximately 7.8% (where patches are marked 'correct' despite failing pre-existing developer tests). Our rigorous full-suite execution eliminates these false positives.
>
> > `Weakness5` Is the 52.2 reported in Table 3, pass@1 or Best@K with test-time scaling?
>
> We have updated Section 3.1 to explicitly clarify this: all reported scores for our models, including the 52.2% in Table 3, are pass@1. We do not employ any test-time scaling or Best@K selection.
>
> ---
>
> [1] R2E-Gym: Procedural Environments and Hybrid Verifiers for Scaling Open-Weights SWE Agents
>
> [2] SWE-smith: Scaling Data for Software Engineering Agents
>
> [3] SWE-rebench: An Automated Pipeline for Task Collection and Decontaminated Evaluation of Software Engineering Agents
>
> [4] SWE-Gym: Training Software Engineering Agents and Verifiers with SWE-Gym
>
> [4] Are “Solved Issues” in SWE-bench Really Solved Correctly? An Empirical Study
>
> ---

---

### Official Review · Reviewer_LqMo · 2025-11-01

**Soundness:** 3
**Presentation:** 3
**Contribution:** 3
**Rating:** 6
**Confidence:** 2

**Summary:**

The paper targets the scaling bottleneck of building verifiable issue-resolution datasets. Authors introduce SWE-Mirror, a pipeline that mirrors real GitHub issues from one repo into another repo that already has a configured Gym. Authors further release a dataset, SWE-MIRROR-60K, has 60k validated tasks across 40 repos and 4 languages. Post-training Qwen2.5-Coder (7B/32B) on collected agent trajectories improves results on SWE-Bench-Verified and Multi-SWE-Bench-Flash.

**Strengths:**

- Authors address an important bottleneck (building verifiable Gym environments) in SWE benchmarks:.
- The dataset is useful for the community and the LLMs trained on that show strong empirical results.
- The paper is easy to follow.

**Weaknesses:**

- It is unclear how the method ensures there is no data contamination or overlap with existing SWE-Bench repositories. I think clarifying how mirrored issues are checked against known evaluation repositories would strengthen the paper.

**Questions:**

1. Is the post-training phase in Section 3.1 based on supervised fine-tuning using the collected trajectories?

---

> ### Author Response · Authors · 2025-11-28
> **Author Response**
>
> We sincerely appreciate your thorough feedback. We address your comments and describe our revisions below.
> > `Weakness1`: It is unclear how the method ensures there is no data contamination or overlap with existing SWE-Bench repositories. I think clarifying how mirrored issues are checked against known evaluation repositories would strengthen the paper.
>
> **Answer1**: We have updated the Ethics Statement section to about the contamination issue. We ensure no data contamination with existing evaluation repositories: all issues and gyms related to SWE-Bench and Multi-SWE-Bench repositories have been carefully filtered out from our SWE-Mirror dataset.
>
> ---
>
> `Question1` Is the post-training phase in Section 3.1 based on supervised fine-tuning using the collected trajectories?
>
> **Answer2**: Detailed description of our training settings and algorithm are expanded in the Agentic Post-training paraphrase in Section 3.1, detailed observation of different training algorithm are discussed in our `observation2` in Section 3.3.1. We use SFT with our presented *Error Masking* to train our models with collected trajectories. Detailed processes are discussed in Agent *Trajectory Collection*.

---

### Official Review · Reviewer_sqN8 · 2025-11-02

**Soundness:** 3
**Presentation:** 3
**Contribution:** 4
**Rating:** 6
**Confidence:** 4

**Summary:**

The paper targets a very concrete bottleneck in software-engineering (SWE) agent research: creating large, verifiable, real-world–like issue-resolving tasks without having to set up a new Gym / execution environment for every single GitHub issue. Current approaches either (i) synthesize problems inside a small number of Gyms (high scale, lower realism) or (ii) automate Gym setup for real issues (high realism, very high engineering/storage cost). To address this, the authors propose to “mirror” an issue/PR from a source repo into a target repo that already has a Gym. For that, they:

1) Collect real, “mirrorable” PRs/issues for each target codebase using GitHub search + LLM filtering.
2) Mirror the issue in three LLM-driven steps: (i) generate a test.patch that would detect the issue in the target repo; (ii) generate a mirror.patch that actually injects the issue in the target repo; (iii) generate a problem statement consistent with SWE-bench style; and
3) Verify with three runs (mirrored, mirrored+tests, mirrored+tests+fix) and strict transition rules (effective tests, effective fix, no regressions, no flakiness). Only tasks that pass the filters are kept.
This results in a dataset with 60,671 validated tasks across 40 repos / 4 languages, with deduplication on F2P tests and patch content; Python is the majority but non-Python is non-trivial.

Following, they train Qwen2.5-Coder-Instruct-7B and -32B on \~12k agent trajectories (~ 6k newly generated by strong proprietary LMs on SWE-MIRROR + ~6k from SWE-rebench). They evaluate on SWE-Bench-Verified and Multi-SWE-Bench-Flash, outperforming several comparable open-source baselines and showing a clear scaling curve w.r.t. number of trajectories and that their error masking training strategy is better than “response only” or “error pruning”.

The contribution is 3-fold:
1) a new paradigm for scaling SWE datasets (mirror instead of synthesize or setup new gym),
2) a large dataset of 60K+ verifiable tasks, and
3) empirical evidence that this data actually makes open models better at agentic coding and follows scaling laws.

**Strengths:**

- The paper is very explicit about the real blocker: not lack of real issues, but lack of reusable, configured, verifiable environments. This is aligned with recent SWE-rebench / SWE-Gym / Multi-SWE-bench discussions, so the motivation is strong and timely.
- The proposed idea is conceptually neat and have a potential for big impact in the code agents community, by finding a path between building more gyms and creating synthetic issues.
- Large dataset (60,671 tasks) with strong emphasis on verifiability: Size is comparable to or bigger than other recent SWE agent datasets, but the authors keep the verifiable property (hidden tests, pass/fail transitions, no regression), which is what the community currently values most for training & RL. This clearly increases the practical value of the release. This answers the usual criticism of LLM-generated SWE data (“is it actually executable?”).
- Non-Python coverage and cross-lingual experiment. Most current SWE agent work is Python-only. The paper actually shows that training on Rust/Go/JS still helps Python SWE-Bench-Verified, which is a nice, non-obvious result and makes the dataset more generally useful.
- Informative ablations show the dataset is not just large, it is useful at increasing sizes, and the proposed error-masking training strategy wins more as data grows.

**Weaknesses:**

- Claim about “new SOTA among Qwen2.5-Coder-Instruct–based LLMs” is slightly narrow. In Table 3, the proposed model (52.2%) is far above base Qwen2.5-Coder-32B (6.2%), so that part is solid. But the same table shows Qwen3-Coder at 69.6% under OpenHands, i.e. a newer base family already higher. The paper should make its claim narrower and explicit (“…for models trained from Qwen2.5-Coder-Instruct, not for newer Qwen3 models”). It seemed too strong the claim of SOTA results.
- Related to that, evaluation table mixes unlike baselines. On Table 3, the proposed SWE-MIRROR-LM-32B (52.2%) is compared to Qwen3-Coder (69.6%) and to very recent large systems under possibly different scaffolds. The main text says “we established a new SOTA among Qwen2.5-Coder-Instruct based LLMs on OpenHands”, which is true, but the table, as written, can be misread as “we’re competitive with the best open models”, which is not the case. The paper should separate clearly: (i) within-family comparison (Qwen2.5-Coder-Instruct), (ii) same scaffold comparison, and (iii) head-to-head with strongest open models.

**Questions:**

- It would be interesting to have a broader discussion, even some categorization, on the cases the pipeline failed to mirror. We may lose important cases there, no?
- Can you explain better the reasoning behind the choice of models as baselines on table 3? Are them with the same backbone model? Qwen2.5-Coder. Do they allow me to compare how good is your dataset compared to other gyms?
- Is there a risk of data contamination on your data?
- You could discuss a bit the licensing of the code you are using, on the ethics statement.

Typos:
- (line 251, table 2 legend) “stastics” -> “statistics”
- (line 226) “must fixes somethings”

---

> ### Author Response · Authors · 2025-11-28
> **Author Response [1/2]**
>
> We sincerely thank you for your careful comments and thorough understanding of our paper. Here we give point-by-point responses to your comments and describe the revisions we made to address them.
>
> ---
>
> > `Weakness1 & Weakness2`: Claim about “new SOTA among Qwen2.5-Coder-Instruct–based LLMs” is slightly narrow. In Table 3, the proposed model (52.2%) is far above base Qwen2.5-Coder-32B (6.2%), so that part is solid. But the same table shows Qwen3-Coder at 69.6% under OpenHands, i.e. a newer base family already higher. The paper should make its claim narrower and explicit (“…for models trained from Qwen2.5-Coder-Instruct, not for newer Qwen3 models”). It seemed too strong the claim of SOTA results.... Related to that, evaluation table mixes unlike baselines. On Table 3, the proposed SWE-MIRROR-LM-32B (52.2%) is compared to Qwen3-Coder (69.6%) and to very recent large systems under possibly different scaffolds. The main text says “we established a new SOTA among Qwen2.5-Coder-Instruct based LLMs on OpenHands”, which is true, but the table, as written, can be misread as “we’re competitive with the best open models”, which is not the case. The paper should separate clearly: (i) within-family comparison (Qwen2.5-Coder-Instruct), (ii) same scaffold comparison, and (iii) head-to-head with strongest open models.
>
> **Answer1**: We agree that the comparison should be more nuanced to accurately reflect the landscape of current LLMs. We have made following updates to address your concern.
> 1. *Refined the claim to be strictly scope-specific*. We have changed the claim to: "We achieve state-of-the-art performance among models fine-tuned from the Qwen2.5-Coder-Instruct (32B) family." This ensures the claim matches the exact contribution—improving a specific open-weight base model—without overstating its position against the absolute strongest systems.
> 2. *Restructured Table 3 to separate unlike baselines*. To prevent the "misread" mentioned by the reviewer, we have reorganized the evaluation table (Table 3) into three distinct sections. This separation ensures readers can distinguish between methodological effectiveness and absolute model capability:
>     - (i) Within-Family Comparison: (e.g., Base Qwen2.5-Coder-Instruct based models). This highlights the specific gain from our method.
>     - (ii) Same-Scaffold Comparison: Comparing against other open models running strictly under the OpenHands scaffold.
>     - (iii) Reference SOTA: Stronger/Newer models (e.g., Qwen3-Coder, Claude 3.5) are now listed in a separate grayed-out "Reference" block. This provides context on the upper bound of current capabilities without implying a direct "competitive" parity.
> Overall, we have revised the text and tables to strictly reflect these bounded claims, ensuring the presentation is transparent and fair. We sincerely hope this reorganization addresses your concerns.

---

> ### Author Response · Authors · 2025-11-28
> **Authore Response [2/2]**
>
> > `Question1` It would be interesting to have a broader discussion, even some categorization, on the cases the pipeline failed to mirror. We may lose important cases there, no?
>
> **Answer2**: As detailed in Appendix A.3 and Table 6, we have analyzed 400 sampled attempts to categorize failure cases, which fall into two primary buckets:
> 1. Compile/Syntax Errors (18%): Dominant in compiled languages like Rust (36%) and Go (28%). These failures imply we likely lose tasks involving strict type constraints or intricate build dependencies that are difficult for the model to satisfy without full project context.
> 2. Semantic Errors (36%):  These occur when the issue's logic is not easily portable—meaning the bug relies heavily on specific, non-abstractable states of the source repository or requires deeper understanding of the  larger edits to the target repository, which our current two-stage workflow struggles to handle effectively at once.
> Improving the success rate of our mirroring process to capture these more complex cases is a primary focus of our future work to reduce the lost of some real-world issues.
>
> ---
>
> > `Question2`: Can you explain better the reasoning behind the choice of models as baselines on table 3? Are them with the same backbone model? Qwen2.5-Coder. Do they allow me to compare how good is your dataset compared to other gyms?
>
> **Answer3**: We selected the Qwen2.5-Coder-Instruct series as our backbone to align with recent state-of-the-art methods (e.g., SWE-Gym, SWE-Smith), which also use this family of models. This ensures our results in Table 3 are directly comparable to the leading open-source efforts in the field.
>
> To specifically address your question about comparing dataset quality across different "Gyms" without confounding variables (like training hyperparameters or teacher models), we conducted a new strictly controlled study. We trained the same backbone model using identical settings. You can view the experiment setting and results in our answer2 to reviewer#hCtU. As shown in results, SWE-Mirror demonstrates competitive quality with a 10.8% resolve rate—virtually matching real-world data (SWE-rebench: 11.4%) and outperforming synthetic baselines—while achieving superior efficiency by generating ~1,516 tasks per Gym (vs. ~266 for SWE-Smith and 1 for SWE-rebench), validating its ability to scale high-quality training data within a limited set of environments.
>
> ---
>
> > `Question3 & Question4` Is there a risk of data contamination on your data...You could discuss a bit the licensing of the code you are using, on the ethics statement.
>
> **Answer 4**: We have included contaimination and licencing issues in Ethic Statement section. To prevent contamination, we explicitly filtered out all repositories and issue instances included in the SWE-Bench and Multi-SWE-Bench evaluation sets during our data collection phase. Regarding licensing, we ensured that all data was sourced exclusively from public GitHub repositories with permissive licenses (e.g., MIT, Apache 2.0) that authorize reuse and modification.

---

### Meta-Review · Area_Chair_Tn12 · 2026-01-07

**Summary:**

SWE-MIRROR is a pipeline that transforms real-world GitHub issues into verifiable tasks using existing Gym environments. Applied to 40 repositories, it yields SWE-MIRROR-60K with 60K+ tasks. Models trained on it show substantial improvements, new SOTA on SWE-Bench-Verified, strong scaling, effective Error Masking, and cross-lingual generalization.

**Reviewer Concerns:**

（1）SOTA claim: Too broad; Qwen3-Coder outperforms the reported 52.2%, and Table 3 mixes baselines across different model families and scaffolds. （2）Evaluation gaps: Lacks comparison with recent SWE-scaling methods (SWE-Smith, R2E-Gym); unclear how synthetic data quality compares to prior approaches. （3）Data concerns: Potential contamination with existing SWE-Bench repositories is not addressed; licensing and ethical issues of mirrored code are missing. （4）Dataset diversity: Low repository diversity may limit generalization; ablation studies use small subsets only. （5）Method clarification: Training protocol, task-mirroring order, context length, and metric definitions (pass@1 vs Best@K) need explanation.

After the rebuttal, I think several concerns exisit, regarding baselines comparion,  dataset diversity.

**Reviewer Scores:**

The reviewers would keep their score unchanged

---

### Decision · Program_Chairs · 2026-01-26

Reject